# TourRank: Utilizing Large Language Models for Documents Ranking with a Tournament-Inspired Strategy

## Abstract

Large Language Models (LLMs) are increasingly employed in zero-shot documents ranking, yielding commendable results. However, several significant challenges still persist in LLMs for ranking: (1) LLMs are constrained by limited input length, precluding them from processing a large number of documents simultaneously; (2) The output document sequence is influenced by the input order of documents, resulting in inconsistent ranking outcomes; (3) Achieving a balance between cost and ranking performance is challenging. To tackle these issues, we introduce a novel documents ranking method called TourRank [1], which is inspired by the sport tournaments, such as FIFA World Cup. Specifically, we 1) overcome the limitation in input length and reduce the ranking latency by incorporating a multi-stage grouping strategy similar to the parallel group stage of sport tournaments; 2) improve the ranking performance and robustness to input orders by using a points system to ensemble multiple ranking results. We test TourRank with different LLMs on the TREC DL datasets and the BEIR benchmark. The experimental results demonstrate that TourRank delivers state-of-the-art performance at a modest cost.

## CCS Concepts

• **Information systems → Language models**.

## Keywords

Large Language Model for Search; Zero-Shot Ranking

**ACM Reference Format:**
Anonymous Author(s). 2018. TourRank: Utilizing Large Language Models for Documents Ranking with a Tournament-Inspired Strategy. In *Proceedings of Make sure to enter the correct conference title from your rights confirmation emai (Conference acronym 'XX).* ACM, New York, NY, USA, 15 pages. https://doi.org/XXXXXXX.XXXXXXX

## 1 Introduction

Recently, Large Language Models (LLMs) have demonstrated great potential in numerous Natural Language Processing (NLP) tasks, especially under the zero-shot settings. Researchers and practitioners have also tried to leverage LLMs document ranking, a core task in information retrieval, under the zero-shot settings. Most of the existing LLM-based document ranking methods can be divided into three

[1]Anonymous code can be seen on https://anonymous.4open.science/r/TourRank-E891.

categories: the *Pointwise* approach that prompts LLMs to independetly assess the relevance of each candidate document [7, 10, 24, 31]; the *Pairwise* approach that use LLMs to compare each document against all the other documents [21]; and the *Listwise* approach that instruct LLMs to generate a ranked list of document labels according to their relevance to the query [13, 19, 20, 25, 33].

While these three approaches lead to different trade-offs between effectiveness and efficiency, the listwise approach, such as RankGPT [25], is considered as the preferred prompting strategy for the LLM-based zero-shot document ranking task. Unlike the pointwise approach, the listwise approach considers multiple documents simultaneously and thus yields better effectiveness in ranking. Meanwhile, listwise ranking eludes the quadratic growing cost of comparing every pair of documents in the candidate list, resulting in improved efficiency than the pairwise approach.

Although the listwise approaches achieve a good trade-off between effectiveness and efficiency and thus are considered preferred prompting strategies for LLM-based document ranking, they also face certain challenges: (1) The maximum context length of LLMs limits the number of documents that can be compared in a single prompt; (2) The listwise generation process can not run in parallel, which makes it hard to return the final ranking list under a tight time constraint. (3) The ranking results are highly dependent on the initial order of the candidate documents in the input prompt.

To address these challenges, we need to develop a prompting strategy for LLM-based document ranking that can: (Requirement 1) establish a global ranking for about 100 candidate documents through multiple local comparisons of 2 to 10 documents in a single prompt; (Requirement 2) parallelize multiple LLM inferences to minimize the overall ranking time; and (Requirement 3) effectively leverages the initial order of candidate documents set by the first-stage retrieval model without becoming overly dependent on it.

Interestingly, we find that using LLMs and prompts to rank documents for a query can be analogous to ranking teams or athletes in a sports tournament, as the design of a sports tournament has similar requirements. A tournament in sports is a structured competition involving multiple teams or individual competitors who compete against each other in a series of matches or games, with the goal of determining a champion or ranking the participants. Figure 1 shows the format and results of an example tournament, the 1982 FIFA World Cup. The tournament consists of two group stages and two knockout stages (i.e., the semi-finals and the final). Analogous to Requirement 1, each group in the group stages and each two-team match in the knockout stages served as a local comparison; the results of these local matches determined which teams could advance to the next stage and their final rankings in the tournament. To expedite the ranking process, the World Cup organized multiple parallel matches across different groups. This parallelization allowed the tournament to progress efficiently and fit into a tight 4-week schedule, which meets Requirement 2. Regarding

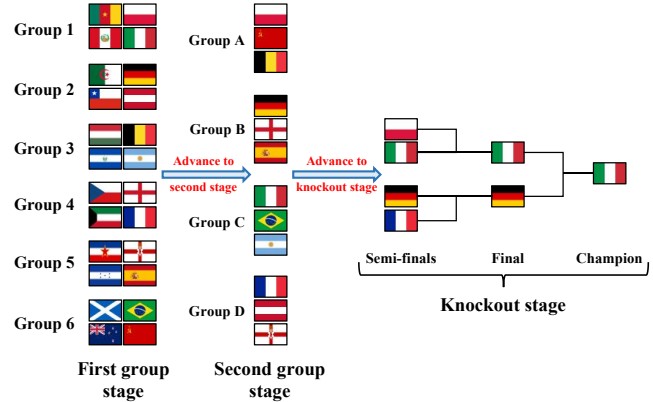

**Figure 1: The 1982 FIFA World Cup.**
In the first group stage, 24 teams were divided into six groups, and the top 2 out of 4 teams in each group qualified. In the second group stage, 12 teams were divided into 4 groups, and only the top 1 out of 3 teams in each group advanced. In knockout stages, only the winner in each two-team match progressed to the next stage.

Requirement 3, the initial groupings were based on seeding and previous performance, providing an initial order of teams. However, the tournament did not solely rely on these seedings; each team's performance in the group stage and subsequent rounds determined their advancement and final rankings.

Therefore, inspired by the tournament mechanism, we propose a new zero-shot document ranking method called **TourRank**, which can fulfill the three requirements and mitigate the challenges in existing methods. In TourRank, we regard each candidate document as a participant in a multi-stage tournament. In each stage, we group the candidate documents and prompt the LLM to select the most relevant documents in each group to advance to the next stage. The LLM inferences across different groups in a single stage can be parallelized. We also design a grouping strategy, similar to the seeding strategy in sports tournaments, to make use of the initial document order provided by the first-stage retrieval model in ranking. In addition, to further improve the effectiveness and robustness, we design a points system to assign different points to each candidate document based on its ranking in each round tournament and perform multiple rounds of tournament. In this way, we can ensemble the results in each round of tournament into a single ranking list based on the final accumulated points in descending order.

To demonstrate the effectiveness of our approach, We test Tour-Rank and baselines on the TREC DL 19 [4], TREC DL 20 datasets [3], and 8 datasets from BEIR benchmark [28]. TourRank achieves state-of-the-art performance on the TREC DL datasets and the most tasks of BEIR benchmark, and achieves a good balance between performance and resource consumption. Experiments on different retriever and initial orderings demonstrate the robustness of TourRank ranking. We further evaluate TourRank using a range of large language models (LLMs), including gpt-3.5-turbo, gpt-4-turbo, gpt-4o-mini via OpenAI's API, as well as several open-source models, such as Mistral-7B-Instruct-v0.2 [9], Llama-3-8B-Instruct

[14] and vicuna-13b-v1.5 [30]. The results suggest that TourRank consistently outperform some existing listwise ranking approaches.

## 2 Related Works

With the development of pre-trained language models like BERT [5] and T5 [22], researchers have leverage them in document ranking. Notably, Nogueira and Cho [15] develop a multi-stage text ranking system using BERT, while Nogueira et al. [16] and Zhuang et al. [32] employ T5 for document ranking. With the emergence of large language model (LLM), recent studies have utilized LLMs for ranking tasks, employing pointwise, pairwise, and listwise approaches. Pointwise methods, such as Query Generation (QG) [24] and Binary Relevance Generation (B-RG) [10], use LLMs to compute the probability or likelihood of query-passage pairs. Pairwise approaches, such as Pairwise Ranking Prompting (PRP) [21], leverage LLMs to conduct pairwise comparisons and ranking of retrieved documents. Luo et al. [12] propose PRP-Graph which utilizes a scoring Pairwise Ranking Prompting unit to construct a ranking graph and aggregates it to enhance LLMs in re-ranking tasks. RankGPT [25] is a listwise method that adopts a sliding window strategy for document ranking. Setwise prompting [33] enhances efficiency by reducing model inferences and prompt token consumption. ListT5 [29] is a reranking approach that uses Fusion-in-Decoder architecture and tournament sort for efficiency and effectiveness, and we specifically clarify the differences between ListT5 and our approach in Appendix A.2.

More introduction of existing works can be seen in Appendix A.

## 3 Method: TourRank

In this section, we introduce a novel zero-shot ranking approach called TourRank, which is inspired by the tournament mechanism. To overcome the limited input length of LLMs and improve the ranking speed, we propose parallel multi-tournaments with multi-stage grouping. And the independent accumulated points system helps TourRank achieve a faster and more robust ranking.

Next, we first delineate how a basic tournament works in Tour-Rank. Then, we explain how to get the accumulated points of the candidate documents, which are subsequently utilized for document ranking. Lastly, we propose a specific grouping method to circumvent the constraints on the input length of LLMs and make full use of the initial ranking order.

### 3.1 A Basic Tournament

For each query, the TourRank approach runs $R$ rounds of tournaments to rank $N_1$ candidate documents retrieved by the first-stage retrieval model. In one tournament of TourRank, we select $N_K$ documents from $N_1$ candidates in a process that consists of $K$ sequential stages and each document gets a corresponding point after a whole tournament. As shown in Figure 2 (a), we choose the documents by stagewise selection ($N_1 \rightarrow N_2 \rightarrow \cdots \rightarrow N_{K-1} \rightarrow N_K$). In the $k$-th selection stage ($k \in \{1, 2, \cdots K-1\}$), the top-$N_{k+1}$ documents to the given query are selected from $N_k$ documents to next selection stage. We add 1 point to each document whenever it is selected to advance to the next stage. In this way, after a full round of tournament, all candidate documents can get the corresponding points. As shown in Figure 2 (a) and Table 1, the $N_k - N_{k+1}$ documents that have

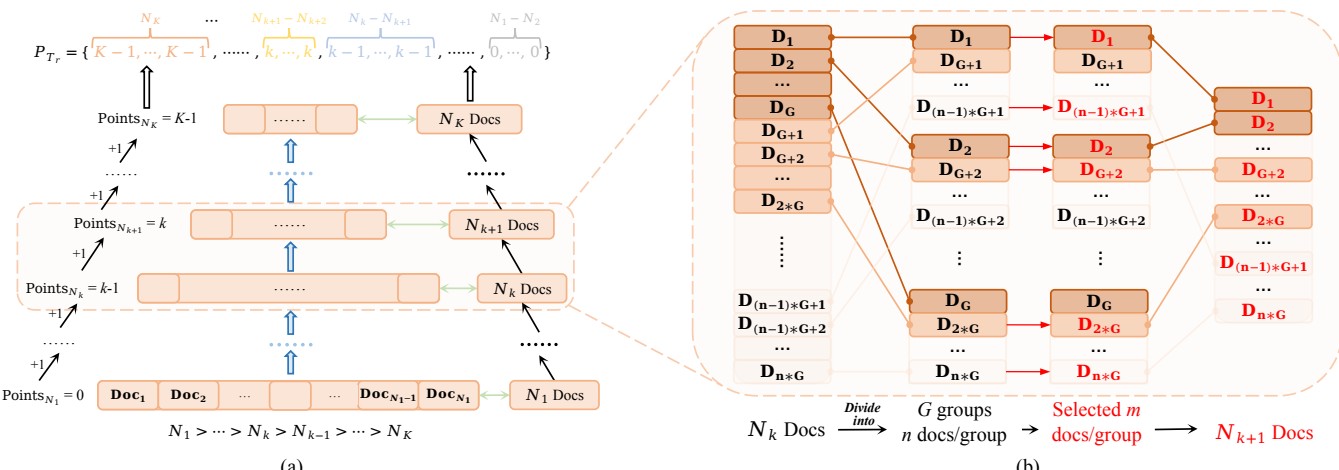

(a)                                                                                          (b)

Figure 2: (a) A basic tournament that selects the $N_k$ most relevant documents from $N_1$ candidates with $K$ stages. $P_{T_r}$ is the points vector for all candidates obtained in the $K$ stages. (b) The grouping strategy in the selection stage of the tournament.

qualified for the $k$-th stage but fails to advance will get $k - 1$ points. As a special case, the $N_k$ most relevant documents that champion in the $K$-stage tournament will get $K - 1$ points. We denote the points vector obtained in the $r$-th round of the tournaments as $P_{T_r}$.

| Number of Docs | Points of Docs |
|:---:|:---:|
| $N_K$ | $K - 1$ |
| $N_{K-1} - N_K$ | $K - 2$ |
| $\cdots$ | $\cdots$ |
| $N_k - N_{k+1}$ | $k - 1$ |
| $\cdots$ | $\cdots$ |
| $N_1 - N_2$ | $0$ |

Table 1: The points of all candidate documents after one tournament. For example, there are $N_k - N_{k+1}$ documents with a score of $k - 1$. ($k \in \{1, 2, \cdots K - 1\}$)

## 3.2 Getting The Accumulated Points

Since the points obtained by one tournament are coarse, multiple tournaments are required to obtain more fine-grained document points. Figure 3 illustrates the process of multiple tournaments, where we can see that points of candidate documents $P_{T_r}$ ($r \in \{1, \cdots, R\}$) are obtained after each round of the tournament.

Because there are many factors that affect the output content of LLMs, such as decoding strategy, temperature coefficient, and especially the order of documents input to LLMs, may introduce some bias, so each set of points vectors ($P_{T_1}, \cdots, P_{T_R}$) obtained by $R$ rounds of tournaments are a little bit different. If these points are added up, the variance of each round tournament could be reduced to some extent, and the accumulated points $P_T$, which is expressed as Equation (1), are more fine-grained and robust. So the final ranking list is obtained according to the accumulated points

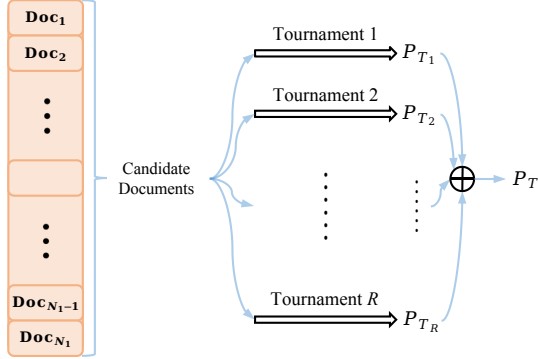

Figure 3: Get the accumulated points of all candidate documents through $R$ tournaments.

$P_T$ in descending order. The analysis in Appendix C shows how TourRank-$r$ improves document ranking.

$$P_T = \sum_{r=1}^{R} P_{T_r} \tag{1}$$

## 3.3 The Grouping and Selection Strategy

Considering the limitation of the input length of LLMs, in some stages of TourRank, such as the stage of selecting $N_{k+1}$ documents from $N_k$ candidates in Figure 2 (a), we may not be able to input all $N_k$ documents into LLMs at once. Therefore, we take the approach of assigning $N_k$ candidate documents to several groups and then parallelly prompt LLMs to select top relevant documents within each group, respectively. Such a grouping strategy is similar to the group stage in a sport tournament.

As shown in Figure 2 (b), the $N_k$ documents are divided into $G$ groups, each of which contains $n$ documents. Here the relative order of $N_k$ initial documents is given by the retrieval model, such as

BM25 [23], etc. When grouping in a sports tournament, the seeded players and the weaker players are evenly assigned into different groups to ensure the fairness of the competition. Similarly, we used a similar strategy to group the documents by evenly distributing the documents in the initial order into different groups as shown in Figure 2 (b). In this way, there will be some difference in the relevance of the documents within a group, making it easier for LLMs to select the more relevant documents.

Additionally, Liu et al. [11] find that current language models do not robustly access and use information in long input contexts because of the position bias. In order to eliminate the bias of LLMs on document input order and achieve a robust ranking, the order of documents in each group will be shuffled before entering LLMs and the multiple tournaments will be performed as shown in Figure 3.

After grouping the documents, we select the most relevant $m$ documents from the $n$ ($m < n$) documents in each group. In Figure 2 (b), we mark the selected $m$ documents in red in each group, and these documents advance to the next stage.

Eventually, through the $k$-th selection stage of the tournament, $N_{k+1}$ more relevant documents are selected from the $N_k$ documents to advance to the next stage. Benefiting from this smart grouping stage and multi-round tournaments mechanism, we solve the problem of limited input length of LLMs while achieving a more robust selection.

### 3.4 The Overall of TourRank

---

**Algorithm 1** The Pseudo-code of TourRank

---

1: **Input: The query $q$ and candidate documents list $D$**
2: *Perform $R$ tournaments **in parallel**, $r \in \{1, \cdots, R\}$:*
3:    *Initialize the points as $P_{T_r} = 0$ for $N_1$ documents.*
4:    *Perform $k$-th selection stages, for $k$ in range$(1, K)$:*
5:       *Assign $N_k$ documents to $G$ groups and each group has $n$ documents.*
6:       *Select $m$ documents that are more relevant to the query $q$ from $n$ in each group **in parallel**.*
7:       *Get the selected $N_{k+1}$ documents to advance to next stage.*
8:       *The points $P_{T_r}$ of the selected $N_{k+1}$ documents add 1.*
9:    *Get a set of points $P_{T_r}$ for all $N_1$ documents.*
10: *After $R$ times parallel tournaments, the final points $P_T$ can be obtained according to Equation (1).*
11: *Rank the candidate documents $D$ according to $P_T$ in descending order.*
12: **Output: A re-ranked list of candidate documents $D_{ranked}$**

---

As the Pseudo-code of TourRank shown in **Algorithm 1**, we perform $R$ parallel tournaments as the process in Figure 3 for the given query $q$ and the candidate documents list $D$. In $r$-th round tournament, we first initialize the points of all $N_1$ candidate documents, that is $P_{T_r} = 0$ for $N_1$ documents. Then, we select and increase the points of the documents in a stage-by-stage way in which $K - 1$ times selection stages are executed serially, and this is corresponds to Figure 2 (a). In $k$-th selection stage, we adopt a suitable grouping approach (Figure 2 (b)) to get the $N_{k+1}$ documents which can advance to the next selection stage, while adding

points to the selected $N_{k+1}$ documents. After $R$ rounds tournament, the points $P_{T_r}, r \in \{1, \cdots, R\}$ can be obtained. We can calculate the final points $P_T$ according to Equation (1). Finally, we re-rank the candidate documents list according to the final points $P_T$ in descending order.

The specific hyperparameters of TourRank can be seen in Table 8 in the Appendix F.

## 4 Experiments

Our experiments mainly focus on the following research questions:

- **RQ.1**: How does TourRank perform in ranking tasks?
- **RQ.2**: How robust is TourRank to the first-stage retrieval models and the initial orders of candidate documents?
- **RQ.3**: What is the trade-off between ranking effectiveness and computational cost/efficiency when using TourRank for document ranking?
- **RQ.4**: How does TourRank perform when using different LLMs, including both open-source models and closed API-based models?

### 4.1 Experimental Settings

*4.1.1 Datasets.* We conduct experiments to answer the above research questions on TREC DL datasets [3, 4] and BEIR benchmark [28]. **TREC** is a widely used benchmark in IR research. We use the test sets of TREC DL 19 and TREC DL 20, which contain 43 and 54 queries. **BEIR** is a heterogeneous zero-shot evaluation benchmark. Following Sun et al. [25], we select 8 datasets for evaluation, including Covid, Touche, DBPedia, SciFact, Signal, News, Robust04, and NFCorpus.

*4.1.2 Metrics.* In the next evaluations, we re-rank the top-100 documents retrieved by the first-stage retrieval model. If not specified, we use BM25 as the default retrieval model and PySerini for implementation.[2] We use NDCG@{5, 10, 20} as evaluation metrics.

*4.1.3 Baselines.* We compare TourRank with several state-of-the-art baselines in documents ranking, including the supervised methods:

- **monoBERT** [15]: A ranking method with a cross-encoder architecture based on BERT-large, trained on MS MARCO.
- **monoT5** [16]: A ranking method that calculates the scores using T5 model.

And the zero-shot methods based on LLMs:

- **DIRECT(0, 10)** [7]: A pointwise method which gives the relevance scores ranging from 0 to 10 to each query-document pair in text format using LLMs. Then, rank the documents according to these scores in descending order.
- **Binary Relevance Generation (B-RG)** [10]: A pointwise method which ranks the candidate documents according to the likelihood of "Yes or No" on a query-document pair.
- **PRP** [21]: A pairwise method that reduces the burden on LLMs by using a technique called Pairwise Ranking Prompting.
- **Setwise** [33]: A listwise method that improves the efficiency of LLM-based zero-shot ranking. The authors introduce two Setwise

---

[2]https://github.com/castorini/pyserini

methods, Setwise.bubblesort and Setwise.heapsort. We reproduce the performance of Setwise.heapsort and Setwise.bubblesort based on the code publicly available on Github [3] in the original Setwise paper. The Setwise paper mentioned that the hyperparameter $c = 10$ is the best value for gpt-3.5-turbo API, so we also use $c = 10$ in our experiments. Here $c$ refers to the number of documents compared in a prompt.

- **RankGPT** [25]: A listwise method that uses a sliding window strategy to achieve listwise ranking based on LLMs. The experiments are also based on the code publicly available in the original paper [4].

## 4.2  Experimental Results

**Results on TREC DL datasets**    Table 2 shows the performance of different methods on TREC DL datasets. We compare NDCG@{5, 10, 20}, and the best top-4 results of zero-shot LLM methods are shaded. We reproduce all zero-shot LLM methods with gpt-3.5-turbo API. From the results, we can make the following findings:

**(1)** Our TourRank-10 outperforms all zero-shot ranking baselines. It is worth noting that after two tournaments (TourRank-2) the performance is much better than one tournament (TourRank-1), and TourRank-2 can significantly outperform RankGPT and Setwise. This indicates that TourRank can achieve good results with fewer tournaments.

**(2)** Generally, the two pointwise methods, DIRECT(0, 1) and B-RG, tend to underperform in comparison to the pairwise and listwise methods. This is because the pointwise methods evaluate each document independently to determine whether it is relevant to the query, while pairwise and listwise methods compare each document to several other documents simultaneously.

**(3)** PRP-Allpair achieves about the same performance as Setwise.heapsort (c=10) and RankGPT on TREC DL 19, and outperforms RankGPT on TREC DL 20. However, PRP-Allpair achieve relatively good results at the cost of much higher complexity and resource consumption than RankGPT and TourRank. We discuss the effectiveness and cost of them in Section 4.5.

**(4)** TourRank-10 achieves comparable results to the best supervised methods on TREC DL 19, and on TREC DL 20 TourRank-10 outperforms the best performing supervised method monoT5 (3B). It can be seen that TourRank is the only zero-shot method based on gpt-3.5-turbo API that can do this. We also perform TourRank-$r$ with other close and open-source LLMs in Section 4.6.

**Results on BEIR benchmark**    Table 3 shows the NDCG@10 of different methods on 8 tasks of BEIR benchmark. Due to the cost limitation, we compare two zero-shot LLM methods, RankGPT and TourRank-$r$, on BEIR benchmark. The following are some valuable discussions:

**(1)** TourRank-10 achieves the best performance in 6 out of 8 tasks and the best average NDCG@10 across 8 tasks among zero-shot LLM methods.

**(2)** The average of TourRank-2 (49.46) outperforms RankGPT (49.37) in terms of NDCG@10 in Table 3, which, together with the better performance of TourRank-2 over RankGPT on TREC DL datasets

---

[3]https://github.com/ielab/llm-rankers

[4]https://github.com/sunnweiwei/RankGPT

shown in Table 2, prove that our TourRank algorithm can achieve good results with only a few times tournaments.

**(3)** Note that on the Touche task and Signal task, all supervised methods and zero-shot methods in Table 3 perform even worse than BM25. The NDCG@10 of all methods on these two tasks is low, only about 0.3. According to Thakur et al. [27], the poor performance of neural retrieval models is mainly due to the large number of short texts and unlabeled texts in the Touche dataset.

The results on TREC datasets and BEIR benchmark jointly answer the **RQ.1**.

## 4.3  Sensitivity Analysis to Initial Ranking

We compare 3 different initial ranking: 1) **BM25**: Get top-100 documents by BM25; 2) **RandomBM25**: Shuffle the order of BM25; 3) **InverseBM25**: Reverse the order of BM25. Figure 4 shows the results of RankGPT and our TourRank based on 3 initial rankings and all these experiments are based on gpt-3.5-turbo API.

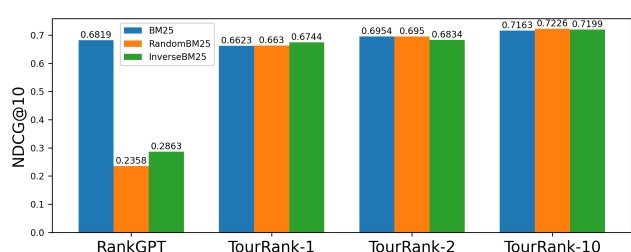

**(a) TREC DL 19**

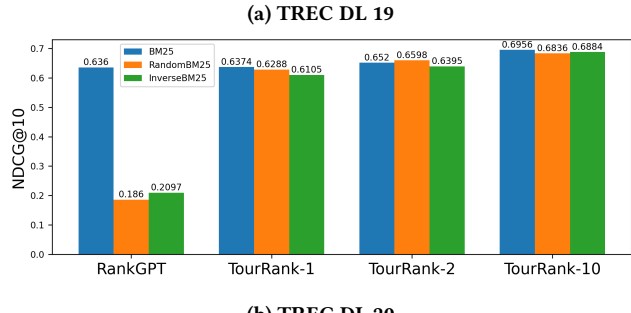

**(b) TREC DL 20**

**Figure 4: The sensitivity analysis to initial ranking of TourRank and RankGPT on TREC DL 19 and TREC DL 20.**

From Figure 4, we can see that RankGPT is very sensitive to the initial permutation of documents list. When the initial permutation is shuffled or reversed, the performance of RankGPT becomes much worse. This is caused by the ranking mechanism of RankGPT, which adjusts the overall permutation of documents list through the sliding window strategy. Sliding the window from bottom to top makes it easier for documents that are originally near the top to be ranked at top positions in the final permutation. Whereas documents that are at the bottom of the initial permutation need to be ranked at the top of every comparison in corresponding sliding window in order to be ranked at the top of the final permutation, otherwise they are left at the bottom or middle of the whole documents list. So, this is the reason why RankGPT is very sensitive to the initial ranking.

| Methods | TREC DL 19 | | | TREC DL 20 | | |
|---|---|---|---|---|---|---|
| | NDCG@5 | NDCG@10 | NDCG@20 | NDCG@5 | NDCG@10 | NDCG@20 |
| BM25 | 52.78 | 50.58 | 49.14 | 50.67 | 47.96 | 47.21 |
| *Supervised Methods* | | | | | | |
| monoBERT (340M) | 73.25 | 70.50 | - | 70.74 | 67.28 | - |
| monoT5 (220M) | **73.77** | 71.48 | - | 69.40 | 66.99 | - |
| monoT5 (3B) | 73.74 | **71.83** | - | **72.32** | **68.89** | - |
| *Zero-Shot LLM Methods* | | | | | | |
| DIRECT(0, 10) | 54.22 | 54.59 | 54.15 | 55.17 | 55.35 | 54.73 |
| B-RG | 63.33 | 62.51 | 60.00 | 65.04 | 63.37 | 60.47 |
| PRP-Allpair | 70.43 | 68.18 | 64.61 | 69.75 | 66.40 | 64.03 |
| Setwise.heapsort (c=10) | 70.55 | 68.16 | 65.63 | 57.05 | 53.73 | 51.66 |
| Setwise.bubblesort (c=10) | 67.62 | 66.19 | 63.41 | 57.03 | 53.82 | 50.79 |
| RankGPT | 72.05 | 68.19 | 62.21 | 67.25 | 63.60 | 59.12 |
| TourRank-1 | 70.95 | 66.23 | 62.49 | 66.65 | 63.74 | 60.59 |
| TourRank-2 | 72.24 | 69.54 | 65.03 | 67.65 | 65.20 | 62.78 |
| TourRank-10 | **73.83** | **71.63** | **68.37** | **72.49** | **69.56** | **66.13** |

Table 2: Performance comparison of different methods on TREC datasets. We reproduce all the zero-shot LLM methods with gpt-3.5-turbo API. The best-performing algorithms for supervised methods and zero-shot LLM methods are bolded, respectively. The best top-4 results of zero-shot LLM methods are shaded in each metric. TourRank-$r$ represents that we perform $r$ times tournaments.

| Methods | Covid | NFCorpus | Touche | DBPedia | SciFact | Signal | News | Robust04 | Average |
|---|---|---|---|---|---|---|---|---|---|
| BM25 | 59.47 | 30.75 | **44.22** | 31.80 | 67.89 | **33.05** | 39.52 | 40.70 | 43.42 |
| *Supervised Methods* | | | | | | | | | |
| monoBERT (340M) | 70.01 | 36.88 | 31.75 | 41.87 | 71.36 | 31.44 | 44.62 | 49.35 | 47.16 |
| monoT5 (220M) | 78.34 | 37.38 | 30.82 | 42.42 | 73.40 | 31.67 | 46.83 | 51.72 | 49.07 |
| monoT5 (3B) | **80.71** | **38.97** | 32.41 | **44.45** | **76.57** | 32.55 | **48.49** | **56.71** | **51.36** |
| *Zero-Shot LLM Methods* | | | | | | | | | |
| RankGPT | 76.67 | 35.62 | 36.18 | 44.47 | 70.43 | 32.12 | 48.85 | 50.62 | 49.37 |
| TourRank-1 | 77.17 | 36.35 | 29.38 | 40.62 | 69.27 | 29.79 | 46.41 | 52.70 | 47.71 |
| TourRank-2 | 79.85 | 36.95 | 30.58 | 41.95 | 71.91 | 31.02 | 48.13 | 55.27 | 49.46 |
| TourRank-10 | **82.59** | **37.99** | 29.98 | **44.64** | **72.17** | 30.83 | **51.46** | **57.87** | **50.94** |

Table 3: Performance (NDCG@10) comparison of different methods on BEIR benchmark. The best-performing algorithms for supervised methods and zero-shot LLM methods are bolded. TourRank-$r$ represents that we perform $r$ times tournaments.

However, our TourRank is quite robust to different initial orderings, as shown by the fact that shuffling and reversing the initial order has almost no effect on TourRank-$r$. The robustness of Tour-Rank to the initial ranking benefits from the tournament mechanism presented in Figure 2. Each tournament is a selection over all candidate documents, not just a fine-tuning of the initial ranking like RankGPT.

## 4.4 Analysis to Different Retrieval Models

In addition to BM25, we also obtain top-100 documents based on two more powerful retrieval models, including a dense retriever model Contriever [8] and a neural sparse retrieval model SPLADE++ ED [6], as the first-stage retrieval model. Then, we perform TourRank and RankGPT to re-rank the top-100 candidate documents retrieved

by different retrieval models based on gpt-3.5-turbo API. The results in Table 4 show that TourRank-10 achieves SOTA ranking performance based on 3 kinds of different top-100 initial candidate documents. And TourRank-2 can also outperform RankGPT in general.

The results in Table 4 and the analysis in Section 4.3 jointly answer the **RQ.2**, that is, TourRank has the ability of robust ranking.

## 4.5 The Trade-Off between Effectiveness and Resource Consumption

In order to prove that our TourRank can achieve a good balance between effectiveness and efficiency, we conduct theoretical analysis and comparison of actual cost and latency.

| Methods | Top-100 | TREC DL 19 | TREC DL 20 |
|---------|---------|------------|------------|
| BM25 | - | 50.58 | 47.96 |
| RankGPT | | 68.19 | 63.60 |
| TourRank-2 | BM25 | 69.54 | 65.20 |
| TourRank-10 | | **71.63** | **69.56** |
| Contriever | - | 62.02 | 63.42 |
| RankGPT | | 69.70 | 68.47 |
| TourRank-2 | Contriever | 69.12 | 71.89 |
| TourRank-10 | | **70.77** | **73.19** |
| SPLADE++ ED | - | 73.08 | 71.97 |
| RankGPT | | 74.56 | 70.75 |
| TourRank-2 | SPLADE++ ED | 74.86 | 74.11 |
| TourRank-10 | | **75.35** | **77.09** |

**Table 4: NDCG@10 of TourRank and RankGPT based on different retrieval models. Here we use gpt-3.5-turbo API for TourRank and RankGPT.**

*4.5.1 Theoretical analysis of efficiency.* Table 5 shows the approximation of the theoretical lowest time complexity of different methods and the number of documents LLMs need to receive. All the contents of Table 5 are based on the recommended parameters. More detailed discussions on precise time complexity and number of input documents are in Table 7 in the Appendix D. From Table 5, we can see that:

| Methods | Time Complexity | No. of Docs to LLMs |
|---------|-----------------|---------------------|
| PointWise | $O(1)$ | $N$ |
| PRP-Allpair | $O(1)$ | $N^2 - N$ |
| Setwise.bubblesort | $\approx O(\frac{1}{9}k*N)$ | $\approx \frac{10}{9}k*N$ |
| Setwise.heapsort | $\approx O(k*log_{10}N)$ | $\approx 10k*log_{10}N$ |
| RankGPT | $\approx O(\frac{1}{10}*N)$ | $\approx 2*N$ |
| TourRank-$r$ | $O(K-1)$ | $\approx 2r*N$ |

**Table 5: A approximation of the theoretical lowest time complexity of various methods and the number of documents which are inputted to LLMs for each method. $N$ is the number of candidate documents. Setwise ranks the top-$k$ ($k < N$) documents through bubblesort and heapsort, and $c = 10$ is the documents compared in a prompt of Setwise based on gpt-3.5-turbo API. $K - 1$ is the times of the selection stages in a tournament (Figure 2 (a)) and $r$ is the times of tournaments in TourRank-$r$. (Note: The approximate contents in this table are based on the recommended parameters.)**

**(1)** Pointwise has the lowest time complexity and the lowest number of documents received by LLMs, but the experimental results of DIRECT(0, 10) and B-RG in Table 2 show that PointWise exhibits poor performance.
**(2)** Although the pairwise method, PRP-Allpair, performs well in the experiments on TREC datasets, the number of input documents required by PRP-AllPair is $N^2 - N$, which will greatly increase

the cost of ranking. And the theoretical optimal time complexity of pairwise method is $O(1)$ when all pairwise documents can be compared at the same time. However, for example, in order to sort 100 candidate documents and achieve the theoretical optimal $O(1)$ time complexity, $\frac{100 \times 99}{2} = 4950$ document pairs should be compared at the same time. While API request rates are often limited in reality, so the actual latency of pairwise method is much higher. We will discuss the actual latency in Section 4.5.2.
**(3)** Setwise performs relatively well on TREC DL 19 dataset in Table 2 and can finish the ranking process with inputting fewer documents to LLM comparing to pairwise method. However, the multiple steps of Setwise have dependencies and cannot be run in parallel, resulting in the high time complexity and resource consumption. And We discuss the actual cost and latency in Section 4.5.2.
**(4)** Two listwise methods RankGPT and our TourRank take into account both the time complexity and the number of documents inputted to LLMs. The experimental results in Table 2 and 3 show that TourRank-2 can outperform RankGPT. From Table 5, we can see that TourRank-2 ($r = 2$) achieves this goal with about twice as many documents to LLMs as RankGPT but with lower time complexity. We also compare TourRank with running RankGPT multiple iterations in serial (Appendix E), and TourRank demonstrates better performance and lower consumption.

*4.5.2 Empirical results for ranking cost and latency.* We compare the actual cost and latency per query of our TourRank-$r$ and several baselines on TREC DL 19 dataset at an API request rate of 30 times per second using gpt-3.5-turbo. Figure 5 provides an intuitive illustration of the relationship between cost, latency, and performance across various methods:
**(1)** The pointwise method B-RG has the lowest actual cost and latency, but NDCG@10 of B-RG is far lower than other methods. It shows that the pointwise method consumes less resources, but it comes at the cost of the worst performance.
**(2)** Although PRP-Allpair perform well in TREC DL datasets in Table 2, the cost and the latency is extremely high comparing to all the other method in Figure 5. In other words, the pairwise method achieves good results with higher resource consumption, which is relatively low cost performance.
**(3)** In addition, we can see that RankGPT and Setwise.heapsort have similar performance as PRP-Allpair, but the cost and latency are much lower than PRP-Allpair. Moreover, the latency of Setwise.heapsort is nearly three times that of RankGPT, which indicates that RankGPT does a better balance between resource consumption and performance than Setwise.heapsort, and Setwise.heapsort is much better than PRP-Allpair.
**(4)** For our TourRank-$r$, when $r = 2$, the NDCG@10 of TourRank-2 has surpassed other baselines, and the cost of TourRank-2 is similar to RankGPT and Setwise.heapsort. The latency of TourRank-$r$ is lower than all baselines except pointwise method, and when $r = 10$, the NDCG@10 of TourRank-10 is significantly higher than all baselines. That is, TourRank-$r$ can achieve good performance at low cost and especially lower latency.

Therefore, TourRank-$r$ can achieve very good performance with low resource consumption, which means TourRank can achieve a better trade-off between effectiveness and efficiency in practice.

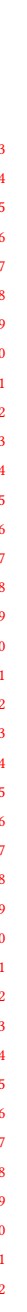

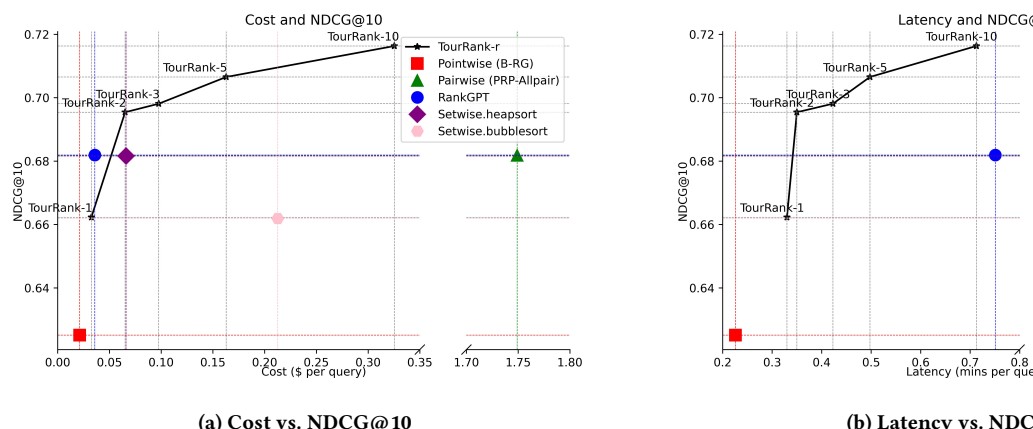

(a) Cost vs. NDCG@10

(b) Latency vs. NDCG@10

Figure 5: Relationship between Cost / Latency and NDCG@10 on TREC DL 19.

The above theoretical analysis and the experimental results of actual cost and latency jointly answer the **RQ.3**.

## 4.6 TourRank Based on Other LLMs

We also explore the effect of zero-shot listwise ranking methods based on different LLMs. Specifically, we perform TourRank, RankGPT and Setwise based on three open-source models Mistral-7B-Instruct-v0.2 [9], Llama-3-8B-Instruct [14], vicuna-13b-v1.5 [30], and OpenAI's API, gpt-4-turbo and gpt-4o-mini.

| Methods | LLMs | TREC DL 19 | TREC DL 20 |
|---|---|---|---|
| BM25 | - | 50.58 | 47.96 |
| RankGPT | | 59.48 | 54.47 |
| Setwise.heapsort | | 58.31 | 44.92 |
| Setwise.bubblesort | Llama-3-8B-Instruct | 49.85 | 34.42 |
| TourRank-2 | | 71.17 | 66.84 |
| TourRank-10 | | **73.30** | **67.25** |
| RankGPT | | 63.90 | **60.80** |
| Setwise.heapsort | | 65.90 | 58.30 |
| Setwise.bubblesort | vicuna-13b-v1.5 | 62.20 | 60.20 |
| TourRank-2 | | 58.55 | 49.37 |
| TourRank-10 | | **66.21** | 59.60 |
| RankGPT | | 61.90 | 58.54 |
| TourRank-2 | Mistral-7B-Instruct-v0.2 | 65.85 | 62.31 |
| TourRank-10 | | **68.64** | **65.04** |
| RankGPT | | 72.67 | 69.48 |
| TourRank-1 | gpt-4-turbo | 72.46 | 67.38 |
| TourRank-5 | | **74.13** | **69.79** |
| RankGPT | | 72.85 | **70.35** |
| TourRank-1 | gpt-4o-mini | 73.35 | 67.89 |
| TourRank-5 | | **75.57** | 70.07 |

**Table 6: NDCG@10 of TourRank and RankGPT based on open-source LLMs, Mistral-7B, Llama-3-8B, vicuna-13b and OpenAI's API, gpt-4-turbo and gpt-4o-mini.**

Table 6 shows the performance of different listwise methods with different LLMs. The top-100 candidate documents are retrieved by BM25. The results of Setwise are still based on the code provided in

the original paper of Setwise. Considering the cost and latency limitations of gpt-4-turbo and gpt-4o-mini APIs, we only run RankGPT as baseline, and TourRank-$r$ only run up to TourRank-5 (r=5).

In Table 6, TourRank-10 still performs very well on the three open-source models, and achieves state-of-the-art performance of in all of them except for TREC DL 20 based on vicuna. Even TourRank-2 sometimes show a relatively good performance. RankGPT's performance on the open-source models is not as good as the gpt-3.5-turbo API, but it still shows a stable and reasonable performance. However, Setwise performs poorly based on Llama3, especially on the TREC DL 20 dataset. In addition, TourRank-5 with gpt-4-turbo or gpt-4o-mini outperforms all methods based on gpt-3.5-turbo in Table 2, which indicates that TourRank can achieve higher performance with fewer tournaments times $r$ based on a stronger model. We also notice that RankGPT performs very well on the stronger models, gpt-4-turbo and gpt-4-mini, and RankGPT is even on par with TourRank-5 on TREC DL 20 based on these two APIs. This shows that RankGPT is also a very good zero-shot documents ranking method based on the strong base model.

In summary, TourRank and RankGPT are more stable and better performing zero-shot document ranking methods.

These experiments shows that TourRank can achieve good performance not only based on OpenAI's API, but also based on open-source LLMs, answering the **RQ.4**.

## 5 Conclusions

We introduce TourRank, a novel zero-shot document ranking method inspired by the tournament mechanism. TourRank allows for multiple tournaments to be conducted in parallel using a multi-process approach. This method effectively addresses challenges faced by large language models in ranking tasks, such as input length limitations, sensitivity to input order, and the difficulty of balancing effectiveness with efficiency.

Furthermore, our experiments indeed demonstrate that TourRank not only outperforms existing LLM-based zero-shot ranking methods but also successfully strikes a good balance between effectiveness and resource consumption.

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

# Appendix

# A  Related Works

## A.1  Neural Network Approaches

Documents ranking has made significant progress, with the help of pre-trained language models, such as BERT [5] and T5 [22]. Nogueira and Cho [15] present a multi-stage text ranking system using BERT, introducing monoBERT and duoBERT models that offer a balance between quality and latency, achieving state-of-the-art results on MS MARCO and TREC CAR datasets. Nogueira et al. [16] introduce a new method for document ranking using a pre-trained sequence-to-sequence model, T5, which outperforms classification-based models, especially in data-poor scenarios, and demonstrates the model's ability to leverage latent knowledge from pretraining for improved performance. Zhuang et al. [32] introduce RankT5, a method for fine-tuning the T5 model for text ranking using ranking losses, which shows significant performance improvements over models fine-tuned with classification losses and demonstrates better zero-shot ranking performance on out-of-domain data.

## A.2  LLMs Approaches

**Pointwise Approaches**    There are several works that employ various zero-shot pointwise rankers. Query Generation (QG) [24] involves rescoring retrieved passages by leveraging a zero-shot question generation model. The model uses a pre-trained language model to compute the probability of the input question, conditioned on a retrieved passage. Binary Relevance Generation (B-RG) [10] proposes to utilize LLMs to make predictions on a query-passage pair, utilizing the likelihood of "Yes/No" responses for the computation of ranking scores. The Rating Scale $0 - k$ Relevance Generation (RS-RG) [31] incorporates fine-grained relevance labels into the prompts for LLM rankers to better differentiate documents of varying relevance levels to the query, thereby achieving more accurate rankings. Guo et al. [7] propose a multi-perspective evaluation criteria-based ranking model to overcome the deficiencies of LLM rankers in standardized comparison and handling complex passages, thereby significantly enhancing the pointwise ranking performance. Guo et al. [7] have also considered the Rating Scale $0 - k$ Directly Score (DIRECT(0, k)) method. This approach prompts the LLM to directly generate the relevance score for each query-passage pair.

**Pairwise Approaches**    Pradeep et al. [18] design a pairwise component to enhance the early precision performance of the text ranking system by employing a pre-trained sequence-to-sequence model (such as T5 [22]) to conduct pairwise comparisons and reranking of retrieved document pairs. Qin et al. [21] introduce a method called Pairwise Ranking Prompting (PRP), which effectively enables LLMs to perform text ranking tasks by simplifying the prompt design and achieving competitive performance across multiple benchmark datasets.

**Listwise Approaches**    LRL [13] enhances text retrieval reranking by employing a large language model as a zero-shot listwise reranker, utilizing a simple instruction template and a sliding window strategy to process multi-document information. Similarly, Sun et al. [25] introduce a novel instructional permutation generation approach called RankGPT, utilizing a sliding window strategy to effectively enable LLMs (such as ChatGPT [17] and GPT-4 [1]) to

be used for relevance ranking tasks in information retrieval, achieving competitive and even superior results on popular IR benchmarks. In addition, both RankVicuna [19] and RankZephyr [20] utilize open-source LLMs and employ instruction fine-tuning to achieve zero-shot listwise document reranking, thereby enhancing the ranking performance of smaller LLMs. Zhuang et al. [33] propose a novel Setwise prompting approach to enhance the efficiency and effectiveness of LLMs in zero-shot document ranking tasks, by reducing the number of model inferences and prompt token consumption, which significantly improves computational efficiency while maintaining high ranking performance. Tang et al. [26] introduces permutation self-consistency, a method to reduce positional bias in large language models for listwise ranking tasks, achieving state-of-the-art performance in passage reranking and sorting datasets. Yoon et al. [29] introduces ListT5, a listwise reranking model that utilizes Fusion-in-Decoder architecture for efficient and robust zero-shot retrieval, outperforming state-of-the-art methods on the BEIR benchmark.

Since both ListT5 [29] and our TourRank both refer to the concept of tournament, it's worth clarifying the difference between two methods: (1) ListT5 is trained on the MS MARCO dataset [2], while TourRank is a completely zero-shot method. (2) The model of ListT5 is trained from an open-source model with encoder-decoder architecture to achieve good ranking efficiency, while TourRank can be applied to both open-source and closed-source models. (3) Although both of two methods propose the basic unit of ranking or sorting inspired by tournament mechanism, the tournament in these two methods are different. In ListT5, the tournament selects the most relevant document. In TourRank, the tournament selects a few most relevant documents, and different stages correspond to different accumulated points $P_{T_r}$ (Table 1). (4) In ListT5, to select the top-k most relevant documents, k times tournaments should be performed sequentially. TourRank, on the other hand, has an independent accumulated points system (Table 1) for each tournament, and multiple tournaments can be executed in parallel, which can significantly speed up the ranking process. The independent accumulated points system is also the key for TourRank to performing multiple tournaments in parallel.

# B  The Performance of TourRank-$r$

Figure 6 shows the trend of NDCG@{5, 10, 20, 30, 50} with the increase of the number of tournaments for TourRank on TREC datasets and BEIR benchmark. We can see that after the first two tournaments, TourRank-2 achieves relatively good results on all datasets, outperforming RankGPT on all corresponding metrics shown. Even in TourRank-10, the metrics still have the potential to continue to increase.

Since the number of tokens consumed by TourRank scales linearly with the number of tournaments, we can control the number of consumed tokens by controlling the number of tournaments. Thus, the balance between effectiveness and token consumption can be achieved.

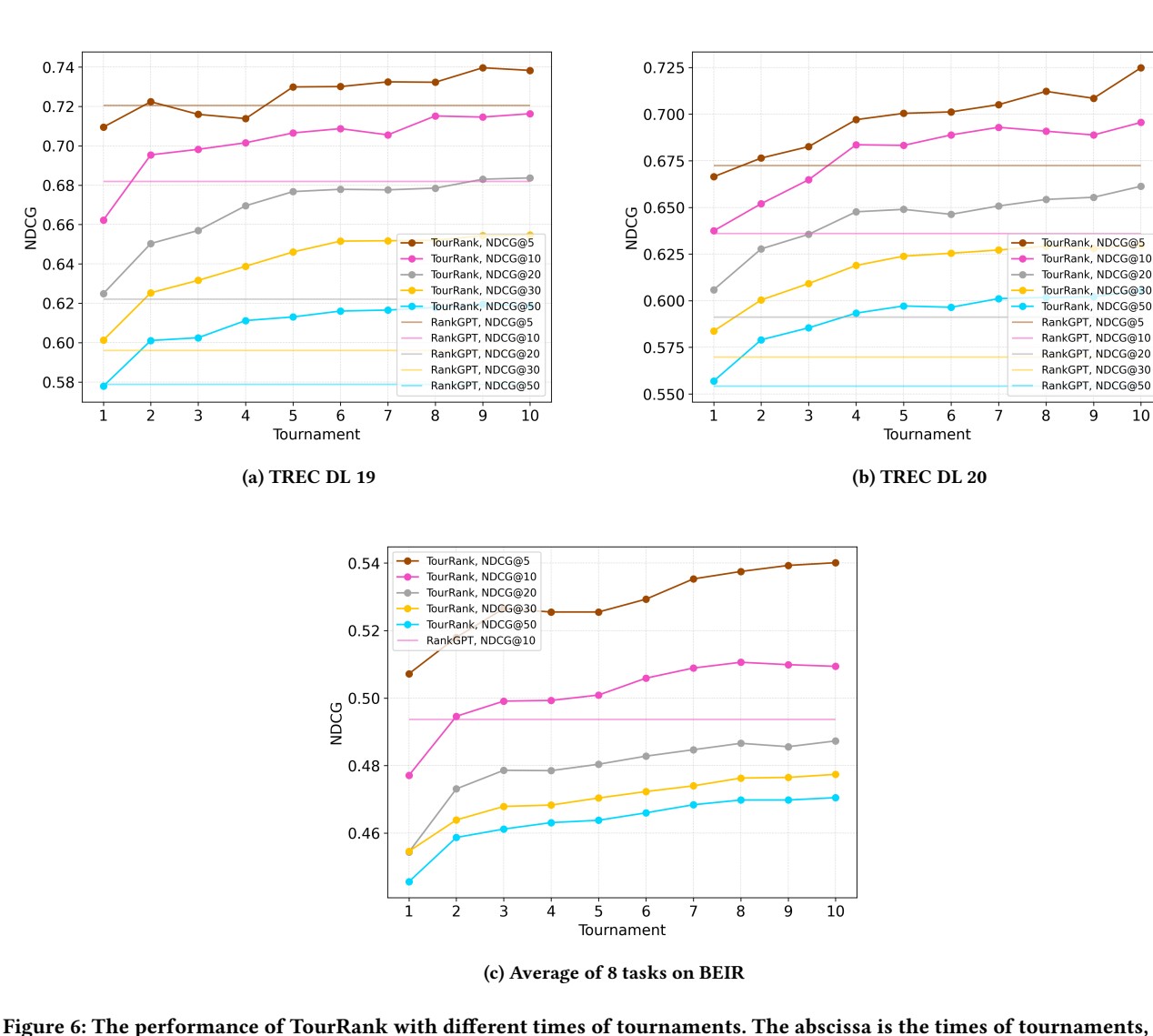

(a) TREC DL 19

(b) TREC DL 20

(c) Average of 8 tasks on BEIR

**Figure 6: The performance of TourRank with different times of tournaments. The abscissa is the times of tournaments, and the ordinate is NDCG@{5, 10, 20, 30, 50}. All the results are based on gpt-3.5-turbo API.**

## C  Case Study: How Does TourRank Improve the Performance of Documents Ranking?

In Figure 7, the horizontal coordinate represents the ranking position of top-50 documents, the red lines represent the accumulated points $P_T$ of TourRank-1 and TourRank-10 respectively, and the blue star points represent the corresponding real labels (integers from 0 to 3).

It can be seen that the $P_T$ of TourRank-1 is coarse, and the labels for the top-50 ranked documents are also relatively scattered. However, the accumulated points $P_T$ of TourRank-10 become much more fine-grained after 10 tournaments, and the labels corresponding to top-50 documents are relatively concentrated. After testing, the NDCG@{10, 50} of the case query have increased from {0.7078, 0.8186} to {0.8715, 0.911}.

Therefore, as the times of tournaments increases, the accumulated points $P_T$ become more fine-grained. This is how exactly TourRank improves the document ranking performance.

## D  The Discussions on Time Complexity and Number of Documents Inputted to LLMs

Table 7 is a more precise version of Table 5 which shows the theoretical lowest time complexity of various methods and the number of documents which are inputted to LLMs for each method. Then, we analysis the content in Table 7.

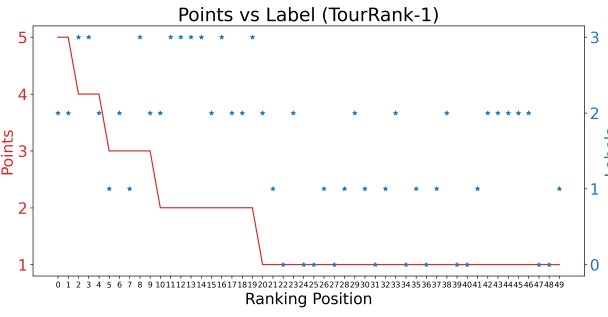
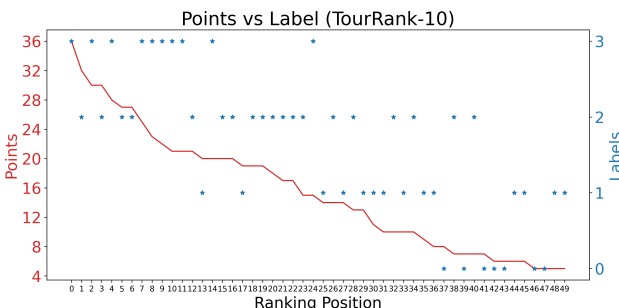

**Figure 7: The relationship between the accumulated points $P_T$ and the corresponding labels for TourRank-1 and TourRank-10. The query of this case is "how long is life cycle of flea" which is one of the queries in the TREC DL 19.**

### D.1 Time Complexity

**PointWise and Pairwise**    Since PointWise scoring a single document and PRP-Allpair comparing a pair of documents can be performed in parallel, the theoretical lowest time complexity is $O(1)$. However, since pairwise methods need to compare about $O(N^2)$ pairs of documents, the theoretical minimum time complexity $O(1)$ is difficult to implement.

**Setwise.bubblesort**    According to [33], the time complexity of Setwise.bubblesort is $O(k * \frac{N}{c-1})$. Setwise rank the top-$k$ ($k < N$) documents through bubblesort, and $c$ is the documents compared in a prompt of Setwise. Considering that Setwise can achieve the best performance with $c = 10$ based on gpt-3.5-turbo API, the time complexity is:

$$O(k * \frac{N}{c-1}) \approx O(\frac{1}{9}k * N)$$

**RankGPT**    RankGPT uses sliding window strategy, so its time complexity is $O(\frac{N-\omega}{s})$. The best window size is $\omega = 20$ and the best step size is $s = 10$ in RankGPT. Based on the optimal parameters ($\omega = 20$ and $s = 10$) and considering that $\omega$ is often much smaller than $N$, the best time complexity of RankGPT is:

$$O(\frac{N-\omega}{s}) = O(\frac{N-20}{10}) \approx O(\frac{1}{10} * N)$$

**TourRank-$r$**    One tournament includes $K - 1$ times selection stages shown in Figure 2, so the time complexity of one tournament is $O(K - 1)$. Because $r$ rounds tournaments can be performed in parallel, the time complexity of TourRank-$r$ is also $O(K - 1)$.

### D.2 No. of Docs to LLMs

**PointWise**    Since the PointWise method scores each document once, the number of documents inputted to LLMs is $N$.

**Pairwise**    However, PRP-Allpair needs to form at least $\frac{N*(N-1)}{2}$ pairs for $N$ candidate documents, and since one pair of documents is inputted to LLMs each time, the number of documents it inputs to LLMs is $N^2 - N$.

**Setwise.bubblesort**    The time complexity of Setwise.bubblesort is $O(k * \frac{N}{c-1})$ and $c = 10$ documents is compared in a prompt, so the number of documents inputted to LLMs for Setwise.bubblesort is:

$$k * \frac{N}{c-1} * c \approx \frac{10}{9}k * N$$

**RankGPT**    In RankGPT, we know that $\omega$ documents need to be inputted into each window, and a total $\frac{N-\omega}{s}$ intra-window ranking need to be performed, so the number of documents input to LLMs is $\omega * \frac{N-\omega}{s}$. The best window size $\omega$ given in RankGPT is 20 and the best step size $s$ is 10. Based on the optimal parameters and considering that $\omega$ is often much smaller than $N$, the number of documents inputted into the LLMs of RankGPT is:

$$\omega * \frac{N-\omega}{s} = 20 * \frac{N-20}{10} \approx 2 * N$$

**TourRank-$r$**    In TourRank, if close to half of the documents are selected to advance to the next selection stage in a tournament (that is, $m \approx \frac{1}{2}n$), the total number of documents input to LLMs is about:

$$N + \frac{N}{2} + \cdots + \frac{N}{2^{K-2}} = \sum_{k=0}^{K-1} \frac{N}{2^k}$$

$$= N * \frac{1 - \left(\frac{1}{2}\right)^{K-1}}{1 - \frac{1}{2}}$$

$$\approx 2 * N$$

The TourRank-$r$ performs $r$ rounds tournaments, so the number of documents inputted to LLMs of TourRank is about:

$$\left(\sum_{k=0}^{K-1} \frac{N}{2^k}\right) * r \approx 2r * N$$

## E    Comparison Between Serial RankGPT and Parallel TourRank-$r$

We also run RankGPT multiple times in seriality called RankGPT (serial), that is, the documents order obtained by this iteration is used as the initial order for the next iteration. Figure 8 shows the comparison of RankGPT (serial) and our TourRank. We can see that on both TREC DL 19 and TREC DL 20 datasets, the NDCG@10 of RankGPT (serial) goes up for the first three iterations, but stops going up after that. This indicates that RankGPT will reach the

| Methods | Time Complexity | No. of Docs to LLMs |
|---|---|---|
| PointWise | $O(1)$ | $N$ |
| PRP-Allpair | $O(1)$ | $N^2 - N$ |
| Setwise.bubblesort | $O(k * \frac{N}{c-1}) \approx O(\frac{1}{9}k * N)$ | $k * \frac{N}{c-1} * c \approx \frac{10}{9}k * N$ |
| Setwise.heapsort | $O(k * log_c N) \approx O(k * log_{10} N)$ | $k * log_c N * c \approx 10k * log_{10} N$ |
| RankGPT | $O(\frac{N-\omega}{s}) \approx O(\frac{1}{10} * N)$ | $\omega * \frac{N-\omega}{s} \approx 2 * N$ |
| TourRank-$r$ | $O(K-1)$ | $\left(\sum_{k=0}^{K-1} \frac{N}{2^k}\right) * r \approx 2r * N$ |

**Table 7: This Table is a more precise version of Table 5. The theoretical lowest time complexity of various methods and the number of documents which are inputted to LLMs for each method. $N$ is the number of candidate documents. Setwise ranks the top-$k$ ($k < N$) documents through bubblesort and heapsort, and $c = 10$ is the documents compared in a prompt of Setwise based on gpt-3.5-turbo API. $\omega = 20$ is window size and $s = 10$ is step size in RankGPT. $K - 1$ is the times of the selection stages in a tournament (Figure 2 (a)) and $r$ is the times of tournaments in TourRank-$r$. All the approximate contents in this table are based on the recommended parameters.**

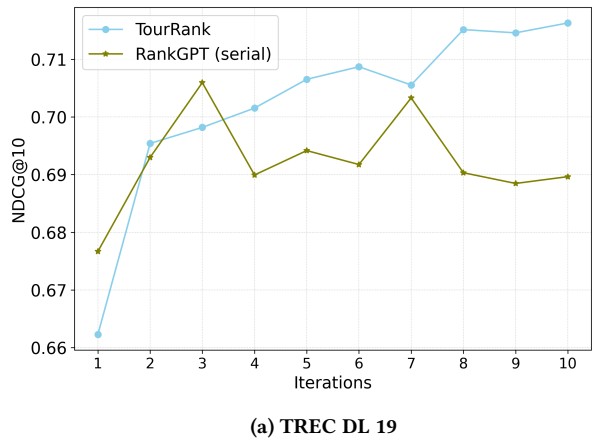

(a) TREC DL 19

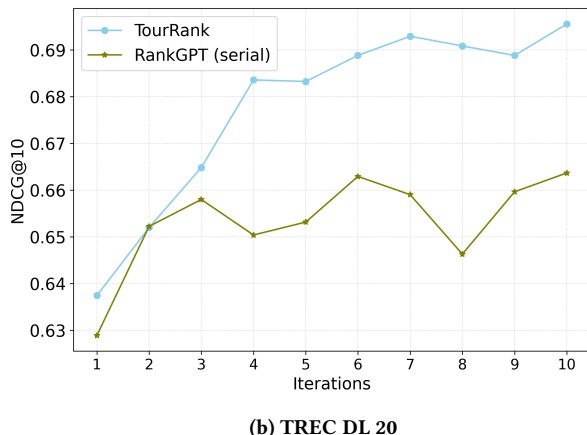

(b) TREC DL 20

**Figure 8: The comparison of NDCG@10 between running RankGPT multiple times in serial and running TourRank-$r$ in parallel.**

upper limit after a few serial runs. However, after multiple iterations (or tournaments) of TourRank-$r$, the NDCG@10 still continues to rise and performs much better than RankGPT (serial).

RankGPT (serial) and TourRank after the same $r$ iterations: (1) The number of documents inputted to LLMs are both about $2r*N$; (2) The time complexity $O(K-1)$ of TourRank is also significantly less than $O(\frac{r}{10} * N)$ of RankGPT (serial); (3) The performance of TourRank is significantly better than RankGPT (serial). These indicate that TourRank can achieve a better balance between effectiveness and efficiency.

## F  The Detail Hyperparameters of TourRank

The detail of hyperparameters of TourRank are shown in Table 8.

Table 9 shows the specific points of candidate document after 1 time tournament under the setting of our experiments.

## G  Prompts

Table 10 shows the prompt used in the grouping and selction stage (Figure 2 (b)) of TourRank.

| Parameters | Explanation | Value |
|:---:|:---|:---:|
| $R$ | The rounds of tournament in TourRank. | 10 |
| $K$ | One tournament contains $K-1$ times selection stages. | 6 |
| $N_k$ | The number of candidate documents in $k$-th selection stages in a tournament. ($k \in \{1, \cdots, K\}$) | $N_1 = 100$ $N_2 = 50$ $N_3 = 20$ $N_4 = 10$ $N_5 = 5$ $N_6 = 2$ |
| $G/n/m$ | $G$: Divide candidate documents into $G$ groups. $n$: Each group has $n$ documents. $m$: Select $m$ documents from each group. | $100 \rightarrow 50 : 5/20/10$ $50 \rightarrow 20 : 5/10/4$ $20 \rightarrow 10 : 1/20/10$ $10 \rightarrow 5 : 1/10/5$ $5 \rightarrow 2 : 1/5/2$ |

**Table 8: Hyperparameters of TourRank.**

| Number of Docs | Points of Docs |
|:---:|:---:|
| $N_6 = 2$ | 5 |
| $N_5 - N_6 = 3$ | 4 |
| $N_4 - N_5 = 5$ | 3 |
| $N_3 - N_4 = 10$ | 2 |
| $N_2 - N_3 = 30$ | 1 |
| $N_1 - N_2 = 50$ | 0 |

**Table 9: The specific points of all documents after one tournament in our experimental settings.**

**system:** You are an intelligent assistant that can compare multiple documents based on their relevancy to the given query.

**user:** I will provide you with the given query and $n$ documents. Consider the content of all the documents comprehensively and select the $m$ documents that are most relevant to the given query: *query*.

**assistant:** Okay, please provide the documents.

**user:** Document 1: $Doc_1$
**assistant:** Received Document 1.

**user:** Document 2: $Doc_2$
**assistant:** Received Document 2.

(User input more documents to assistant.)

**user:** The Query is: *query*. Now, you must output the top $m$ documents that are most relevant to the Query using the following format strictly, and nothing else. Don't output any explanation, just the following format:
Document 3, ..., Document 1

**Table 10: The prompt of the grouping and selection stage of TourRank.**

