# OpenReview forum: "TourRank: Utilizing Large Language Models for Documents Ranking with a Tournament-Inspired Strategy"
_ACM.org/TheWebConf/2025/Conference — WWW 2025 Oral_

### Official Review · Reviewer_qpU3 · 2024-11-25

**Novelty:** 5
**Technical Quality:** 5

**Review:**

This paper introduces TourRank, a novel document ranking strategy inspired by tournament mechanisms, designed to address limitations in using LLMs for zero-shot ranking tasks. Key challenges such as input length constraints, sensitivity to input order, and the trade-off between performance and computational cost are explicitly tackled. The approach employs a multi-stage grouping and selection system, combined with a points-based ranking mechanism. Empirical results on TREC DL datasets and the BEIR benchmark demonstrate the method’s efficacy in achieving robust and cost-efficient performance, outperforming other zero-shot and supervised baselines in many cases.

Pros:
1. The use of a tournament-inspired mechanism for ranking is creative and aligns well with the limitations of LLMs.
2. By incorporating multi-round tournaments and shuffling input orders, TourRank mitigates input sensitivity issues seen in methods like RankGPT.
3. The experiments cover multiple retrieval models, datasets, and metrics, providing comprehensive validation of the method's robustness and adaptability.
4. TourRank achieves state-of-the-art performance on various benchmarks, outperforming comparable methods in most tasks.

Cons:
1. I am concerned that the multi-stage grouping and points accumulation mechanism could introduce additional implementation complexity.
2. Some of the captions and figures are not clear. In Fig.1, why are the captions split into two paragraphs? In Fig.2 (a), the color distinction of text is not clear enough.
3. The writing should be improved.
4. Performance seems to depend significantly on the choice of grouping size, the number of stages, and the number of tournaments.
5. The improvements from increasing the number of tournament rounds might exhibit diminishing returns beyond a certain threshold. TourRank-10 achieves better performance than TourRank-2, but the incremental benefit may not justify the additional resource cost.

**Questions:**

See above

**Reviewer Confidence:**

3: The reviewer is confident but not certain that the evaluation is correct

**Scope:**

3: The work is somewhat relevant to the Web and to the track, and is of narrow interest to a sub-community

---

### Official Review · Reviewer_5WBE · 2024-11-30

**Novelty:** 5
**Technical Quality:** 6

**Review:**

This paper presents TourRank, a novel zero-shot document ranking method inspired by sports tournament mechanisms. The work makes several notable contributions while also having some limitations.

## Pros:
1. Clear problem definition addressing key challenges in LLM-based document ranking: input length limitations, ranking inconsistency, and efficiency-effectiveness trade-off
2. Novel tournament-inspired approach that is well-motivated and clearly explained
3. Comprehensive experimental evaluation across multiple datasets (TREC DL and BEIR benchmark)
4. Strong empirical results showing improvements over existing methods
5. Thorough ablation studies and analysis of different components
6. Good theoretical analysis of complexity and resource consumption

## Cons:
1. The parallel multi-tournament design, while innovative, adds complexity to the system
2. Some parameter choices (e.g., number of tournaments R=10) could use more justification
3. The comparison with ListT5 could be more detailed given their similar tournament inspiration
4. Limited discussion of potential failure cases or limitations
5. Could benefit from more analysis of how different document types affect performance

**Questions:**

1. The key innovation of using grouped and staged tournaments to address input length limitations and ranking inconsistency is interesting. However, the paper lacks sensitivity analysis of the two critical tournament structure parameters (K stages and group sizes). What guidelines would you suggest for selecting these parameters, and how do they impact the method's performance?
2. How does TourRank perform across different document types and lengths? Is there any performance bias towards certain document characteristics? It would be valuable to understand if there are particular document types or lengths where the method performs notably better or worse.
3. In real-world information retrieval systems, document collections often contain hundreds or thousands of documents. How does the method's performance scale with such larger document collections? What are the practical limitations and potential solutions?
4. Given that the method relies on LLMs for ranking, how sensitive is the performance to different prompt designs? Have you explored the impact of various prompting strategies on ranking effectiveness?

**Reviewer Confidence:**

3: The reviewer is confident but not certain that the evaluation is correct

**Scope:**

4: The work is relevant to the Web and to the track, and is of broad interest to the community

---

### Official Review · Reviewer_TwB8 · 2024-12-02

**Novelty:** 6
**Technical Quality:** 4

**Review:**

Please provide an evaluation of the quality, clarity, originality and significance of this work, including a list of its pros and cons (max 200000 characters). Add formatting using Markdown and formulas using LaTeX. For more information see https://openreview.net/faq

This paper addresses the task of prompting Large Language Models (LLMs) to rank documents, introducing a parallel prompting strategy inspired by tournaments in sports. The methodology is empirically validated on TREC DL19, DL20, and BEIR datasets, with additional analyses provided. While the efficiency benefits of the approach are evident and clearly demonstrated, the claimed effectiveness advantages may require more reliable comparision.

Pros:
1. The paper is generally well-organized and easy to follow, with motivations clearly presented through a vivi FIFA example.
2. In addition to the main experiments on TREC and BEIR, the paper provides detailed supplementary analyses, particularly on Time Complexity and Number of Documents to LLMs, which add depth to the study.
3. Partial code is shared in an anonymous repository.

Cons:
1. The supervised baselines appear outdated, as the Mono series is from 2019-2020. For supervised methods, ListT5 from ACL24 has shown better results on TREC DL and BEIR, and the ListT5 paper mentions other recent models outperforming the Mono series. see [1].
2. The baseline setwise performance seems suboptimal; the settings may not be optimized. According to Figure 3(a) in the original setwise paper, the best performance is achieved at c=3, not c=10 as used in Table 2 for Setwise.heapsort and Setwise.bubblesort. Additionally, in Table 2, setwise (c=10) results are notably lower than pointwise B-RG on DL20, which seems inconsistent with the patterns reported in the original setwise paper. see [2]
3. The RankGPT results in the original paper, especially when using GPT-4, achieve 75.59 and 70.56, surpassing the reported metrics for RankGPT in Table 6 of this paper. see [3].
4. For the pointwise method Direct(0, 10). RG-S(0, 4) has better performance. RG-S(0, 4) is also included as a baseline in the paper you cite for Direct(0, 10). see [4, 5]
5. Table 8 details hyperparameter settings, but lacks explanations on how these were chosen or how they might impact the method's effectiveness.

[1] ListT5：ListT5: Listwise Reranking with Fusion-in-Decoder  Improves Zero-shot Retrieval, ACL 24 long
[2] Setwise：A Setwise Approach for Effective and Highly Efficient Zero-shot Ranking with Large Language Models, SIGIR24 long
[3] RankGPT：Is ChatGPT Good at Search?  Investigating Large Language Models as Re-Ranking Agents, EMNLP23 outstanding
[4] Fang Guo, Wenyu Li, Honglei Zhuang, Yun Luo, Yafu Li, Le Yan, and Yue Zhang. 2024. Generating Diverse Criteria On-the-Fly to Improve Point-wise LLM Rankers. arXiv preprint arXiv:2404.11960 (2024).
[5] Beyond Yes and No: Improving Zero-Shot LLM Rankers via Scoring  Fine-Grained Relevance Labels, NAACL 24 short

**Questions:**

See the cons.

**Reviewer Confidence:**

4: The reviewer is certain that the evaluation is correct and very familiar with the relevant literature

**Scope:**

4: The work is relevant to the Web and to the track, and is of broad interest to the community

---

### Official Review · Reviewer_e3zJ · 2024-12-02

**Novelty:** 3
**Technical Quality:** 5

**Review:**

This paper is important for web search and information retrieval. It tackles a key challenge: using large language models (LLMs) to rank documents. Document ranking is a core part of web search, and this work is interesting for both researchers and industry professionals. The methods and experiments in the paper are strong and well-presented.

1. Strengths
 + Clear Explanation of TourRank:
The authors explain the TourRank method clearly. They describe its tournament structure, how it processes tasks in parallel, its scoring system, and how it groups documents.

+ Good Experiments:
The experiments are well-designed. TourRank is compared with other methods on well-known datasets like TREC and BEIR, showing that it works well.

+ Robust Results:
The authors test TourRank with different LLMs, retrieval models, and document orders. This shows that TourRank is reliable in different scenarios. They also analyze the time and cost of using TourRank, which gives useful insights into its efficiency.

2. Practical Focus:
The paper considers real-world issues, like how much time and resources the method needs.

+ Areas for Improvement
Although the paper is strong, there are some areas that could be better:

+ Comparison with Fine-Tuned Models:
The paper only looks at zero-shot methods (no fine-tuning). Comparing TourRank with LLMs that have been fine-tuned for ranking tasks would make the results more complete.

+ Bias in Grouping:
The grouping strategy in TourRank splits documents into groups to help LLMs compare them. But if the initial document rankings are biased, this could affect the results. Testing other grouping methods might help.

+ Scalability:
The experiments only test TourRank with 100 documents. In real-world web search, there are often thousands or millions of documents. Testing TourRank on much larger datasets would show how well it works in real-world scenarios.

**Questions:**

In the section 4.5.2,it does not discuss the impact of K (number of stages per tournament) on computational cost and latency.

**Reviewer Confidence:**

3: The reviewer is confident but not certain that the evaluation is correct

**Scope:**

3: The work is somewhat relevant to the Web and to the track, and is of narrow interest to a sub-community

---

### Official Review · Reviewer_tBsa · 2024-12-02

**Novelty:** 6
**Technical Quality:** 5

**Review:**

This paper proposes TourRank, a zero-shot document ranking method inspired by sports tournament structures. The authors address key challenges in LLMs for ranking, including input length limitations, balancing performance with computational cost and positional bias. TourRank employs a multi-stage grouping strategy, parallel inference, and an ensemble of ranking results based on a tournament-inspired points system. Experiments on TREC DL and BEIR benchmarks show that TourRank outperforms some baselines.

Quality:
The paper is well-grounded in theory. The experiments are comprehensive, comparing TourRank against multiple baselines on common benchmarks.
Clarity:
The paper is clear, easy to read and flows smoothly. The figures are self-explanatory and useful for the discussion.
Originality:
The tournament-inspired approach is novel and convincing.
Significance:
This work is significant for the IR community, especially in scenarios where efficiency and robustness in zero-shot settings are critical.
Pros:
1. A novel tournament-inspired framework that effectively tackles LLM limitations, allowing to maintain a good balance between ranking performance and efficiency.
2. Achieves state-of-the-art results on TREC and BEIR datasets.
3. This method can be adapted to different retrieval models and LLMs, including both proprietary ones.
4. Positional bias of LLMs is mitigated by aggregating scores across multiple rounds with shuffled document orders.
Cons:
1. The reliance on LLMs might hamper applicability in areas with limited computational capabilities or latency constraints (as the latter is measured in the order of minutes).
2. The comparison is limited methods employing LLMs, while related work using simpler approaches (e.g. Pointwise ranking) show better results with lower computational requirements.

**Questions:**

1. How sensitive is the performance of TourRank to the choice of hyperparameters? Are there recommended setups? Are optimal parameters robust to the use of different datasets?
2. Can TourRank be used in scenarios with strict latency requirements?
3. What are the limitations of this approach? Are there any scenarios or datasets where TourRank fails to outperform simpler baselines?

**Reviewer Confidence:**

4: The reviewer is certain that the evaluation is correct and very familiar with the relevant literature

**Scope:**

4: The work is relevant to the Web and to the track, and is of broad interest to the community